

# Enhanced microbial nitrogen transformations in association with intertidal macrobiota

Catherine A. Pfister[1], Mark A. Altabet[2]

[1]Department of Ecology and Evolution, University of Chicago, Chicago IL
[2] School of Marine Sciences, University of Massachusetts, Dartmouth, MA

*Correspondence to*: Catherine A. Pfister (cpfister@uchicago.edu)

keywords: nitrification, nitrate reduction, ammonium regeneration, microbiome, DOC

**Abstract.** Microbial nitrogen processing in direct association with marine animals and seaweeds

is poorly understood. Macrobiota supply a substrate for microbes to reside, and a source of

excreted nitrogen and dissolved organic carbon (DOC). We tested the role of a mussel (*Mytilus

californianus*), a red alga (*Prionitis sternbergii*) and an inert substrate for microbial activity

using enclosed chambers and enriched ammonium and nitrate. Chambers with seawater from the

same environment served as a control. We found that mussels and *Prionitis* elevated ammonium

oxidation and nitrate reduction two orders of magnitude over that of seawater, while the effect of

simply an inert substrate had relatively little effect. Extrapolating to a square meter of shoreline,

microbial activity associated with mussels could oxidize 2.5 mmol of ammonium and reduce per

1.2 mmol of nitrate per day. A square meter of seaweed could produce even higher rates, at

135.2 and 320.5 mmol per day for nitrification and nitrate reduction, respectively. Seawater

collected from the shore versus 2-5 km offshore showed no difference in ammonium oxidation or

nitrate reduction. Microbial nitrogen metabolism associated with mussels did not change whether




we measured it at night or during the day. When we experimentally added DOC (glucose) as a

carbon source, there was no change to nitrification rates. Added DOC did increase DIN and

phosphorus uptake, indicating that elevating the concentration of DOC stimulated heterotrophic

microbial activity, and suggests potential competition for DIN between heterotrophic and

chemolithotrophic microbes and their seaweed hosts. Our results indicate that microbes in direct

association with coastal animals and seaweeds greatly enhance nitrogen processing, and likely

provide a template for a diversity of ecological interactions.

## 1 Introduction

Anthropogenic doubling of the supply of biologically available nitrogen (Galloway et al.,

2008)(Fowler et al., 2013) has increased the importance of understanding the multiple

components of the nitrogen cycle. In marine ecosystems, microbial activity has been shown to be

a key driver in the nitrogen cycle, and while phototrophs can dominate uptake in the water

column (Flombaum et al., 2013), chemolithotrophs and chemoheterotrophs have also been

shown to be quantitatively significant to nitrogen cycling (Capone et al., 2008; Francis et al.,

2007; Zehr and Ward, 2002). In coastal marine areas, the large biomass of macrofauna and

macrophytes presents the opportunity for microbial taxa to form associations where microbes

have habitat as well as a predictable nitrogen supply (Moulton et al., 2016). Many of these

macrobiota are restricted in movement, making them reliable substrates for microbial

populations. Further, animals and plants create strong gradients in oxygen and inorganic and

organic nutrients such that processes that vary over hundreds of meters or kilometers in the open

ocean can change over scales of mm in proximity to an animal (de Goeij et al., 2013) or over a



scale of meters relative to species aggregations (Clasen and Shurin, 2015). There are many

quantitative estimates of microbial nitrogen fluxes, including ammonium oxidation

(nitrification), in seawater from disparate marine locales (Beman, J. Michael et al., 2011; Ward

and Bouskill, 2011). Comparatively, there is little knowledge of the microbially-mediated

nitrogen fluxes associated with nearshore species, including whether the presence of animal and

plant hosts enhance the diversity and/or intensity of microbial functions. With the harvest and

loss of many marine species (Maranger et al., 2008; Worm et al., 2006), the importance of

determining the biogeochemical role of microbes associated with macrobiota becomes more

urgent. Here, we quantify microbial nitrogen processing in coastal and offshore water and in

association with two key coastal species. Because dissolved organic matter is one of the

microbial resources supplied by macrobiota in aquatic systems (Hansell and Carlson, 2015), we

also manipulated dissolved organic carbon (DOC) to examine the effect of carbon availability on

microbial nitrogen processing.

Across diverse aquatic ecosystems, the metabolic activities of animals and plants can generate

the environmental niches necessary for a variety of microbial metabolisms (Allgeier et al., 2014;

Croll, 2005; Layman et al., 2011; Schindler et al., 2001; Subalusky et al., 2015; Vanni, 2002). In

marine systems, there is an increasing appreciation that animals, primarily through their excreta,

contribute significantly to nitrogen supply (Moulton et al., 2016; Pather et al., 2014) which can

help relieve nitrogen limitation of microbial carbon fixation. Microbial nitrogen processing,

including nitrification and nitrate reduction, is enhanced in proximity to animals in marine

systems (Heisterkamp et al., 2013; Pfister et al., 2014, 2016a; Stief, 2013; Welsh and Castadelli,

2004). These enhanced nitrogen metabolisms also contribute to nitrous oxide production

(Heisterkamp et al., 2010, 2013), as well as retention of nitrogen (Pfister et al., 2016a).



The rocky shores of the northeast Pacific are characterized by high levels of dissolved inorganic

nitrogen, both from upwelling and from ammonium excretion by animals., including mammals

(Roman and McCarthy, 2010), birds (Wootton, 1991), and invertebrates (Aquilino et al., 2009;

Bracken, 2004; Pfister et al., 2016a). Open ocean areas are characterized by relatively consistent

gradients in oxygen that generate predictable areas of oxidizing and reducing processes that

might be limited by nutrient concentrations (Bristow et al., 2017). In contrast, high nutrients and

the photosynthetic and respiratory processes of coastal biota can drive wide fluctuations in

oxygen, possibly leading to both oxidizing and reducing microbial metabolisms in the same

location over a diel cycle (e.g, (de Goeij et al., 2013; Pfister et al., 2016b).

A further effect of macrobiota beyond nitrogen regeneration and the production of oxygen

gradients is the production of dissolved organic matter. In addition to ammonium excretion and

dissolved organic nitrogen (DON) production by macrobiota, macroalgae also produce DOC,

likely enhancing select microbial metabolisms. DOC and DON can support different types of

microbial metabolisms in aquatic ecosystems, resulting in divergent outcomes for coastal

productivity and nutrient cycling. DOC release by macroalgae may stimulate heterotrophic

nitrate reduction, where microbes respire DOC with nitrate ($NO_3^-$) or nitrite ($NO_2^-$) as alternative

electron acceptors. DOC can also stimulate the oxidation of $NH_4$ through heterotrophic

nitrification. In addition to promoting microbial transformations between $NH_4$ and $NO_2/NO_3$,

enhancing the DOC supply can result in competition between different microbial metabolisms

for DIN. Work in streams suggests heterotrophic bacteria may compete with chemolithotrophs

for DIN (Butturini and Sabater, 2000), a result that may depend upon the ratio of C:N, where

increasing DOC increases C:N and promotes nitrogen competition (Strauss and Lamberti, 2000).





In sum, increasing the supply of DOC to marine microbes could have counteracting effects on

nitrification rates. While an increase in $NH_4$ oxidation would indicate stimulation of

heterotrophic nitrifiers, a decrease in $NH_4$ oxidation rate would be consistent with increased

competition for $NH_4$ with heterotrophic microbes. While the precise role of DOC in nitrogen

metabolisms is likely varied and still not fully described, DOC contributes greatly to

heterotrophy in microbes and fuels the quantitatively significant 'microbial loop' (Azam, 1998).

The effects that macrobiota have on both nitrogen excretion and DOC release are poorly

understood. We tested how the presence of the California mussel (*Mytilus californianus*), a red

alga (*Prionitis sternbergii*), and the proximity to shore affected microbial nitrogen

transformations during both daylight and nighttime periods. We hypothesized that macrobiota

(both mussels and algae) would enhance microbial nitrogen cycling, and nearshore seawater

would have greater microbial activity compared with offshore because it was in close proximity

to the macrobiota. We used gas-tight chambers and added enriched ammonium ($^{15}NH_4$) or nitrate

($^{15}NO_3$) to estimate the flux of ammonium and nitrate. Further, gas-tight chambers allowed us to

test whether microbial denitrification resulted in loss of nitrogen via $N_2$ gas. We then

manipulated DOC to test its specific effects on nitrogen transformations. In sum, we asked: 1)

how does the presence of mussel, red algal tissue or inert substrates affect microbial nitrogen

cycling?, 2) does the microbial activity in seawater differ between from the nearshore versus 1-5

km offshore?, 3) are there diel cycles in these microbial nitrogen transformations, and 4) does the

experimental addition of dissolved organic carbon (DOC) alter microbial nitrogen metabolism?




## 2 Materials and Methods

### 2.1 Chambers for assaying microbial metabolisms

In order to quantify the microbial nitrogen transformations that both retain and lose dissolved

nitrogen we enclosed seawater and some components of the rocky shore environment within 2

gas-tight Plexiglas chambers. Each 2.26 L chamber measured approximately 15 cm in diameter,

30 cm in height, and contained 2 ports at the top: one for an o-ring sealed connection to an

oxygen probe and the other with a septate lid for gas-tight sampling of seawater. From 29 Jun to

22 Aug 2012, 54 assays were done in the chambers either *in situ* in tidepools 2 km east of Neah

Bay, WA, USA at Second Beach, WA (n=19) (48.23°N, 124.40°W W), at the shore at Tatoosh

Island, WA (n=26) 48.39°N, 124.74°W) or onboard the R/V Clifford Barnes using seawater 2-3

km from each of these shore-based sites (n=9). The Second Beach site is described in Pather et

al., (2014) and has tidepools at a height of 1.2 to 1.5 m above Mean Lower Low Water (MLLW),

with a diversity of species (described in (Pfister, 2007; Pfister et al., 2016b). The chambers were

anchored into a number of these tidepools for 3-5 hours at a time during periods of low tides

when the tidepools were emergent. Thus, the chambers contained tidepool water and were

incubated under natural light and temperature conditions. Experimental trials included tidepool

seawater only (n= 5), seawater with the California mussel *Mytilus californianus* (n=9), or

seawater with bioballs and ceramic rings (n=5). Bioballs are topographically complex 26 mm

plastic balls used in commercial aquaria to provide substrate for microbes, while Filstar™

ceramic rings (1 cm diam) are also used in filtration (Aquatic Eco-systems™); both had been

anchored in the tidepools for one month to initiate a microbial community in which to query

function.



140        At Tatoosh Island, where wave action was more significant, the chambers were placed at

the shore in a water bath within a shaded styrofoam cooler rather than in tidepools. The

chambers were filled with seawater at the shore of Tatoosh Island and contained seawater only

(n=8), seawater with the California mussel *Mytilus californianus* (n=12), seawater with the red

alga *Prionitis sternbergii* (n=3), or seawater with bioballs (n=3). Bioballs had incubated at the

lower edge of the mussel bed for one month prior to use in the chambers. For all experiments, the

wet mass of *Prionitis* was weighed with a Pesola™ spring scale, while the mussel dry mass was

estimated from individual length measurements of the mussels (Wootton, 2004).

        Microbial nitrogen metabolisms were compared in shore-based seawater collections versus

seawater collected offshore in 2012. The offshore samples were collected with a CTD rosette

system with 10 L Niskin bottles on the R/V Clifford Barnes 2-5 km offshore from Tatoosh Island

(48.432°N, 124.73°W) or Second Beach (48.37°N, 124.57°W) at a depth of 1m. The offshore

assays were done with the chambers in a cooler with a water bath onboard the ship deck. We

compared 4 replicates of each shore and offshore chambers during Jun and Jul of 2012.

        We initiated each run by filling the chamber with seawater and any macrobiota or bioballs.

Oxygen and temperature were immediately recorded by a probe that remained in the chamber

through the duration of the experiment. We added an enrichment of either 10000 ‰ of $\delta^{15}NH_4$

(as 0.05M ammonium chloride, $^{15}NH_4Cl$) or 10000 ‰ of $\delta^{15}NO_3$ (as 0.05M sodium nitrate,

$Na^{15}NO_3$, Cambridge Isotopes). We thus increased $^{15}N$-$NH_4^+$ or $^{15}N$-$NO_3^-$ by a factor of ten with

the intention of maximizing our ability to detect the enriched signal in $N_2$ gas. Both ammonium

and nitrate concentrations in seawater in this region is typically high (>2 and >10 $\mu$molL$^{-1}$,

respectively), minimizing any concentration-related effects from tracer addition. The chamber

was agitated to mix the tracer and then agitated 3-4 more times during the 3- to 5-hour



incubation period. No samples were taken during the incubation so that we did not compromise

the gas-tight nature of the chambers. At the end of the incubation, we inserted a needle attached

to a gas-tight syringe through a rubber septa, drew out seawater and injected into a 30 ml serum

vial with a rubber stopper that had been evacuated to 160 mtorr with a Welsh 8905 Vacuum

Pump.

### 2.2 Testing the effects of adding DOC

170        A further set of experiments in 2014 tested whether DOC additions enhanced microbial

nitrogen processing by increasing the concentration of DOC approximately 6 times above the

ambient nearshore concentration to 1000 uM DOC. We added 1.0 ml of a 1.96 M glucose

solution to one chamber at the beginning of the experiment while the other served as a control

across all paired experiments. All paired experimental runs were performed at Tatoosh Island

and resulted in 8 paired runs with seawater, and 4 paired experiments with either bioballs or

*Prionitis*. We used an enrichment target of 2000‰ of $\delta^{15}NH_4$ (as 0.001M ammonium chloride,

$^{15}NH_4Cl$), a decreased enrichment compared to those described above because we were not

trying to detect an enriched signal in $N_2$ gas. This tripling of $^{15}N\text{-}NH_4^+$ allowed us to test whether

ammonium oxidation changed with added DOC; an increase in $NH_4$ oxidation would indicate

stimulation of heterotrophic nitrifiers, while a decrease would be consistent with increased

competition for nitrogen by heterotrophs.

### 2.3 Quantifying enrichment results

In all experiments, a water sample prior to tracer addition ($T_o$) was collected to quantify

concentrations of ammonium, nitrate, nitrite, phosphorus, and silica, as well as natural

abundance isotope levels of $\delta^{15}NH_4$, $\delta^{15}NO_2$, and $\delta^{15}NO_3$. For all target experiments, we used



these initial measures to calculate the exact initial enrichment which could deviate from the

target enrichment due to natural variation in nutrient and $^{15}$N concentration. We collected the $T_o$

sample by filtering ~180 ml of source water through a syringe-filter (Whatman GF/F) into HDPE

bottles, which we kept frozen until analysis. For the final sample ($T_f$) after 3-5 hours of

incubation, we filtered directly from the individual chamber. All nutrient concentrations were

analyzed at the University of Washington Marine Chemistry lab (methods from UNESCO,

1994), while isotope determinations were done at University of Massachusetts, Dartmouth using

methodology for isotopic composition reported previously (Pather et al., 2014; Pfister et al.,

2014, 2016b). Briefly, nitrogen stable isotopes of ammonium were measured according to a

modified version of the $NH_4$ oxidation method detailed in Zhang et al., (2007). $NH_4$ is oxidized

to nitrite using a hypobromite solution and then reduced to $N_2O$ using a sodium azide-acetic acid

reagent before analysis on an IRMS (isotope ratio mass spectrometer). The stable isotope ratios

of nitrate were measured by cadmium reduction to nitrite, followed by reaction with azide to

$N_2O$ (McIlvin and Altabet, 2005). For the DOC analysis, an additional 25 ml were filtered into a

40 ml VOA vial (Shimadzu Inc). We tested for the presence of enriched $N_2$ gas in the chambers

deployed in 2012 using sample collection and analytical procedures described in (Charoenpong

et al., 2014). Chamber oxygen and temperature were recorded with a Hach™ HQ4D and a LDO

probe.

### 2.4 Quantifying microbial transformations

Stable isotope enrichment experiments can quantify nitrogen processing in marine

environments by tracking the transfer of the tracer between its source and product pools (Glibert,

Pamela M. et al., 1982; Lipschultz, 2008). The traditional isotope tracer transfer model generally



involves estimating a single rate parameter from time 0 to time t (Lipschultz, 2008) and has the

general form:

$$Rate = (R_k(t) - R_k(o))/[(R_s(o) - R_k(o)) * \Delta t] * [\bar{k}]$$    Equation (1)

where $k$ is the sink or product at time $t$ (or the average $\bar{k}$), $s$ is the source and $R$ designates

the atom % ($^{15}$N/($^{15}$N +$^{14}$N )x100) of either the source or sink component. The source-product

model (Equation 1), is thus used to estimate individual nitrogen transformation rates. We

estimated ammonium oxidation to nitrite with a $^{15}$NH$_4$ tracer experiment, while monitoring the

$^{15}$N enrichment in nitrite. Nitrate reduction to nitrite was estimated with a $^{15}$NO$_3$ tracer

experiment, while monitoring the $^{15}$N enrichment in nitrite.

A previous study of enrichment in tidepools showed substantial flux in inorganic nitrogen

pools that was best described with differential equation models fit to multiple time points, and

underestimated with source-product models (Pfister et al., 2016b). Source-product models likely

underestimated the oxidation of ammonium here too because remineralization by species within

the chamber diluted the $^{15}$NH$_4$ tracer. We still used the simpler source-product models because

we had only a two-sampling protocol, at the beginning and the end of the experiment, to prevent

gas escape.  Isotope dilution is important and indicates ammonium remineralization by species

within the chamber. We quantified ammonium remineralization in chambers with $^{15}$NH$_4$ tracer

using the methods of (Pather et al., 2014). Briefly by fitting an exponential model decline in

$\delta^{15}$NH$_4$ from the beginning to the end of the experiment y=$ae^{-bx}$. The parameters $a$ and b were

fitted where $b$ was the exponential decay constant in the $\delta^{15}$N$_{NH4}$ enrichment. Remineralization

rates were thus calculated as:

$$NH_4^+ \ remineralization = |-b| * [\overline{NH4}]$$    (Equation 2)

in nmol L$^{-1}$ h$^{-1}$, where $[\overline{NH4}]$ was the mean concentration of ammonium in nM.



## 3 Results

### 3.1 Dynamics of nutrients and isotopes in chambers

The presence of either the California mussel or the red alga *Prionitis* amplified net changes

to ammonium and nitrate concentration in the experimental chambers compared with chambers

that contained bioballs or only coastal seawater (Fig 1). Chambers during daylight hours with

*Prionitis* and mussels had increased ammonium over the course of the experiment compared

with the relatively unchanged coastal seawater and bioball treatments (Fig 1a, $F_{5,51}=6.150$,

$p<0.001$), while nitrate decreased with *Prionitis* and increased with mussels (Fig 1b, $F_{5,51}=3.512$,

$p=0.008$). Changes in nitrite did not differ among treatments (Fig 1c, $F_{5,51}=0.66$, $p=0.659$).

The dynamics of $\delta^{15}N_{NH4}$, $\delta^{15}N_{NO2}$, and $\delta^{15}N_{NO3}$ within the chambers revealed transfer of

$^{15}N$ isotope and thus microbial transformations. When $^{15}N$-$NH_4^+$ was added, enrichment in

$\delta^{15}N_{NO2}$, and $\delta^{15}N_{NO3}$ and any dilution in the $\delta^{15}N_{NH4}$ signal was measured (Figure A1 a, b).

Similarly, enrichment in $\delta^{15}N_{NH4}$ and $\delta^{15}N_{NO2}$ followed the addition of $^{15}N$-$NO_3^-$ (Figure A1 c,

d). Deviations in our target of initial enrichment (10000 and 2000‰) occurred due to natural

variation in nutrient concentrations at the time of tracer addition.

### 3.2 Nitrogen transformation rates

Microbial nitrogen processing rates increased when either the California mussel or the red

alga *Prionitis* was present. Ammonium oxidation rates with the mussel (14.1 nmol $L^{-1}$ $h^{-1}$) or red

alga (32.8 nmol $L^{-1}$ $h^{-1}$) were two orders of magnitude greater than ammonium oxidation in

seawater only or with bioball surfaces which were less than 1 nmol $L^{-1}$ $h^{-1}$ (Fig 1d, $F_{5,25}=15.19$,

$p<0.001$, logged values). Our estimates of nitrate reduction from the addition of $Na^{15}NO_3$ were

also 2 orders of magnitude greater with *Prionitis* (63.7 nmol $L^{-1}$ $h^{-1}$) and mussels (6.7 nmol $L^{-1}$ $h^-$



[1]) compared with bioballs and seawater (Figure 1e, $F_{5,19}$=17.64, p<0.001, logged values). For all

these estimates of microbial nitrogen processing, we found high overlap in the rates estimated

with living, intact mussels compared with mussel shells only, indicating that the responsible

microbes reside on the shell surface, rather than the mussel tissues (Figure 1d, e). The presence

of mussels was further associated with increased ammonium remineralization, remineralization

with mussels was twice that with bioballs and the red alga *Prionitis*, and an order of magnitude

more that seawater alone (Figure 1f).

Our estimates of mussel or algal mass within each chamber resulted in per gram estimates

of the effect of these macrobiota on nitrogen transformation rates. For every gram of mussel dry

mass, 3.21 nmol (se=0.64) of ammonium were oxidized per L per hour, while 1.60 nmol of

nitrate were reduced (se=0.41). A comparable contribution is made per g of *Prionitis* wet mass

with 1.50 nmol ammonium oxidized (se=0.27) and 1.56 nmol nitrate reduced (se=0.72).

DIN uptake in chambers could be due to both microbial transformations or seaweed uptake.

We thus estimated what percentage of total DIN uptake was attributed to microbial activity based

on our tracer enrichment and found that nitrate reduction accounted for as little as 4.2% of the

decrease in nitrate concentration during the day, but as much as 87.2% at night. Estimates of

ammonium oxidation revealed that ammonium oxidation made up 5.2 – 7.4 % of total

ammonium uptake during the day.

**3.3 Day versus night nitrogen transformations**

Nitrification in association with mussels or in seawater alone did not differ between night

and day hours ($F_{1,11}$=0.583, p=0.461), but chambers with mussels again showed ammonium

oxidation rates two orders of magnitude higher than seawater only (Fig 2a). Nitrate reduction

was ten times higher when mussels were present compared to the microbial activity of seawater



only (mussels>seawater, $F_{1,14}=68.1$, p<0.001), with greater daytime rates (day>night, $F_{1,14}=5.83$, p=0.030), suggested only for seawater (e.g a near interaction, $F_{1,14}=3.76$, p=0.073). Seawater nitrate reduction rates during the day (1.15 nmol $L^{-1}$ $h^{-1}$) were four times greater than those at

night (0.26 nmol $L^{-1}$ $h^{-1}$, Fig 2b), though these rates were still small relative to mussels.

### 3.4 Onshore versus offshore microbial nitrogen transformations

The seawater only chambers showed no difference in ammonium oxidation rates whether collected at the shore (mean=0.23 nmol $L^{-1}$ $h^{-1}$) or offshore (mean=1.12, Fig 3a, $t_{3.8}=1.65$,

p=0.177), although there was low sample size (n=4). Overall, there was little change in nutrient concentration when seawater from either offshore or nearshore was isolated; the overall mean change in DIN was less than 1 uM for both nearshore (-0.70) and offshore (0.63). There was also no difference in silica uptake between the two regions either (t=-0.679, p=0.525), indicating that diatom activity did not differ in the two regions.


### 3.5 Nitrogen transformation rates with added DOC

On average, the coastal seawater that was used in the chambers contained 145 uM DOC; replicates with the addition of DOC were increased approximately 6 times that amount to1000 uM. *Prionitis* enhanced the DOC in the chambers also, with a mean increase in DOC of 9.31

mmol C per hour over the course of the experiment (n=4). Nitrification rates did not change significantly when glucose was added (Fig 4a), although we acknowledge that our sample size was small and nitrification was not detected in some instances across both treatments, perhaps impeding a strong test of glucose effects. However, DOC addition did change nutrient uptake rates. The addition of DOC to experimental chambers generally resulted in greater uptake of



nitrite and nitrate with *Prionitis*, bioballs or in seawater alone while ammonium showed a trend

toward greater uptake only with *Prionitis*, otherwise there was little overall change in ammonium

concentration (Fig 4 b,c,d). DOC addition was also associated with an increased uptake of DIN

and phosphorus, regardless of the composition of the chamber (Fig 4 e,f). Silica was unchanged

with bioballs or seawater alone, while there was greater uptake of silica with *Prionitis*,

suggesting *Prionitis* hosts diatoms (Fig 4g).

The greater uptake of DIN in chambers with supplemental DOC could be due to increased

microbial respiration with DOC. The effect of glucose on the uptake of DIN or phosphorus did

not differ based on whether seawater, bioballs or *Prionitis* was in the chamber ($F_{2,31}$=0.645,

p=0.531 and $F_{2,31}$=0.264, p=0.770, respectively), suggesting that the background metabolism of

heterotrophic bacteria was the same regardless of the substrate available and even though the

DIN concentration declined a further 6.5 uM on average when *Prionitis* was present.  If

microbial respiration increased with DOC, we were unable to detect it by measuring oxygen

concentrations. Whether we pooled treatments for seawater, bioballs and *Prionitis* or examined

them separately, dissolved oxygen measurements did not differ (t= 1.125, p=0.277, df=16).


## 4 Discussion

### 4.1 Seascape Scale Importance of Macrobiota for Microbial N Metabolism

The per mass estimates of microbial nitrogen transformations that we measured reveal

significant microbial processing rates along coastal shorelines. Studies from (Wootton, 2004)

estimate that a square meter of mussel bed can contain 32,425 g dry mass of mussel.

Extrapolating from our measurements on both day and night, microbial nitrification in a square

meter of mussel bed would amount to 2.5 mmol per day, with an additional 1.2 mmol of nitrate



reduction. As a comparison, at this site it would take a volume of seawater of 1 million liters to

host the same microbial nitrogen metabolism, the equivalent of a 10 m by 10 m area of the ocean

to 10 m depth.

A similar calculation can be done for macroalgae using (Paine, 2002) control plots in the

intertidal at Tatoosh Island where macroalgal mass was estimated to be 8.6 kg per square meter.

If *Prionitis* has any functional similarity to other seaweeds sampled by (Paine, 2002), then

ammonium oxidation could reach 135.2 mm per day for a square meter of seaweed, while nitrate

reduction would be 320.5 mmol. The macroalga contribution is thus comparable to water column

nitrification only when we consider a volume in excess of 50 million liters, or an area

approximately 71 m on a side to a depth of 10 m. Even if *Prionitis* is exceptional with respect to

microbial function when compared with other seaweeds, the potential contribution of macroalgae

to microbial function could be substantial. Thus, independent of macroalgal effects on DIN

uptake (Fig 1a, b, c) or via ammonium remineralization by mussels (Fig 1f), the microbiome of

each of these species makes distinct contributions to nitrogen cycling.

We demonstrated that seawater isolated from the immediate vicinity of benthic substrates

had similar rates of nitrogen metabolism offshore (Fig 3). As this measurement indicates no

difference in the activity of suspended microbes, we conclude that microbial metabolism was

elevated due to microbes directly associated with the mussel and the red alga. Previous analyses

of 16s rRNA sequencing of mussels and *Prionitis* (Pfister et al., 2014) and metagenomic analysis

of mussel shell microbes (Pfister et al., 2010) both indicate DNA sequences associated with a

diversity of nitrogen metabolisms. The similarity of nearshore to offshore microbial function

would appear at odds with our previous work showing that natural isotopes of ammonium and

nitrate ($\delta^{15}N_{NH4}$ and $\delta^{15}N_{NO3}$) are enriched near the shore, indicating increased microbial



processing. However, the nearshore results likely reflect benthic-associated activity influencing

adjacent seawater.

## 4.2 Microbial Metabolism and Dissolved Organic Carbon

When considering the effect of DOC in microbial assemblages, there are 3 groups of

microbes that might be affected. There are nitrifiers that are either heterotrophic or

chemolithotrophic (Ward, 2008), as well as heterotrophic bacteria that might consume DOC and

assimilate ammonium, but not nitrify (e.g. Kirchman, 1994). Thus, added DOC might be

expected to increase heterotrophic nitrification if DOC was limiting nitrifier growth.

Alternatively, added DOC could decrease nitrification if generalist heterotrophic bacteria were

stimulated and then outcompete chemolithotrophs for ammonium (Butturini and Sabater, 2000),

although we do not know if ammonium was ever limiting. A third possibility is that

heterotrophic nitrifiers are such a small percentage of nitrification activity that there is no

detectable effect of elevated DOC. We found mixed evidence for the effects of DOC on

nitrification. Ammonium oxidation was never stimulated by DOC (Figure 4); if anything, there

was a nonsignificant trend of decreased ammonium oxidation with glucose, suggesting that

general heterotrophic bacteria were consuming the elevated DOC. Our DOC additions were

accompanied by decreased dissolved inorganic nitrogen and phosphorus in the surrounding

seawater, suggesting that heterotrophic microbial metabolism increased, a result consistent with

other glucose addition studies with microbes (Zhang et al., 2013). Bacterial production in

seawater has been shown to increase with glucose addition (Caron et al., 2000; Jacquet et al.,

2002), with heterotrophic bacteria released from carbon limitation when DOC is added (Jacquet

et al., 2002; Joint et al., 2002). In streams, glucose additions have shown decreased nitrification

with DOC (Strauss and Lamberti, 2000), a result attributed to heterotrophic bacteria in direct



competition with nitrifiers. While (Strauss and Lamberti, 2000) documented decreased oxygen

and increased respiration with added DOC, we detected no effect of DOC on the change in

oxygen within chambers (Fig. 4h). The unknown contribution of photosynthesis to oxygen

concentrations, as well as the relatively high oxygen content of the seawater in these locales

could have masked oxygen differences. Nonetheless, DOC stimulated nutrient uptake,

presumably by heterotrophic microbes and the effect of DOC was the same whether seawater,

bioballs or *Prionitis* was in the chamber (Fig 4b-i). Thus, the background metabolism of

heterotrophic bacteria was unchanged even when *Prionitis* was present and reduced chamber

DIN concentrations 6.5 uM over the course of the experimental runs.

A final explanation to explain the increased DIN uptake with added DOC is that bacteria

are able to compete with any phototrophs for nitrate when an organic carbon source is increased

(e.g. (Diner et al., 2016). Nitrate reduction rates are high with *Prionitis* and this alga also

provisions DOC, perhaps promoting the coupling of heterotrophy and nitrate reduction. Whether

any of the decreased nitrate concentration associated with *Prionitis* in chambers could be

attributed to heterotrophic nitrate reduction is unknown at this time, because our experiments

with added DOC did not assay nitrate reduction, only ammonium oxidation.

In sum, while DOC concentrations can be elevated in nearshore areas compared with

offshore, there was little evidence that enhanced DOC changed nitrification rates, even in the

chambers with *Prionitis*, where DIN levels were lower due to seaweed uptake. Whether

heterotrophic nitrifiers are present remains unknown, though previous analysis of microbes at

these sites suggest the presence of taxa associated with heterotrophic nitrification (e.g.

*Arthrobacter* (Hynes and Knowles, 1982), Crenarchaeota (Offre et al., 2013), and *Alcaligenes*

*faecalis*, (Joo et al., 2005), though they were detected in only a small fraction of samples (Pfister



et al., 2014). Analysis of 16s rRNA of seawater, mussels and *Prionitis* do show sharp

distinctions in β−diversity, with some taxa unique to each (Pfister et al., 2014).

Taken together our data suggest that chemolithotrophic nitrifiers are dominating

nitrification in this area. Other heterotrophic bacteria can noticeably depress DIN and phosphate

concentrations when DOC is supplemented, suggesting there may be some carbon limitation for

heterotrophic microbial metabolisms. If, as suggested by (Strauss and Lamberti, 2000), the C:N

ratio in the water column determines the relative fitness of heterotrophic bacteria versus

chemolithotrophic nitrifiers, then the many regions where DIN concentrations in seawater are

lower than they are at our Washington coastal sites may show a different result.

Of note is that many seaweeds produce detectable amounts of DOC in coastal areas (Wada

and Hama, 2013), with as much as 14% of net primary production being released as DOC in a

kelp species (Reed et al., 2015). Among other seaweeds, 20 to 30% of that DOC can be taken up

within 2 hours (Brylinsky, 1977), suggesting an active heterotrophic assemblage in proximity.

Seaweeds also have a diverse assemblage of microbial associates (Lemay et al., 2018; Marzinelli

et al., 2018; Michelou et al., 2013; Pfister et al., 2014). Which of these associated microbes

benefit from this DOC and whether others are inhibited is unknown. While we tested the effect

of elevated glucose on nitrification with enriched ammonium, a next step is to test if those

microbes involved in the nitrate reduction pathways are affected by glucose addition.

Macrobiota that serve as hosts for microbes provide a predictable substrate for attachment

in a fluid environment and provide dissolved organic matter in many forms (Carlson and

Hansell, 2015). The mussels studied here also excrete ammonium and likely DON (Bayne and

Scullard, 1977; Pather et al., 2014). Their filter feeding activities release DOC in many forms,

and continually process organic matter that can be utilized by microbes (Jacobs et al., 2015).



Through filter feeding and mucus production, there is increasing evidence that marine

invertebrates and microbes are connected through their production and use of dissolved organic

matter (Rädecker et al., 2015; Rix et al., 2016).

### 4.3 The multiple factors influencing nitrogen availability

Our experiments provide insight into the fate of nitrogen in coastal systems. While

ammonium oxidation and nitrate reduction rates were two orders of magnitude higher than any

water column estimates, we have no evidence that nitrate reduction continued through to

denitrification and the release of $N_2$ gas as we never detected enriched $^{15}N$ in $N_2$ gas (e.g. (Jensen

et al., 2011). Thus, nitrogen was being retained in our experimental system. If ammonium

oxidation and nitrate reduction are occurring relatively constantly, as suggested by our

experiments, then a diversity of microbially-mediated DIN dynamics may take place across

microenvironments that differ in oxygen levels. The net result could be continued microbial use

of ammonium and nitrate and the ability for the microenvironment surrounding the animal or

seaweed to sustain a range of microbial metabolisms, a result obtained for other marine

invertebrates (de Goeij et al., 2013; Heisterkamp et al., 2013). Research in tidepools containing

these same species has also shown both nitrogen oxidation and reduction processes (Pfister et al.,

2016b). In all instances to date, the metabolism of the host macrobiota results in a daily range of

oxygen levels, thus providing a diversity of environmental niches that favor different microbial

transformations through time.

**5 Conclusions**

The marine mussel and alga species studied here were loci for microbial nitrogen

metabolism, elevating ammonium oxidation and nitrate reduction two orders of magnitude over



that of seawater alone. For mussels, microbial nitrogen processing did not differ between

daylight and nighttime hours. While the addition of DOC did not increase ammonium oxidation,

it resulted in greater uptake of DIN, suggesting that DOC stimulated heterotrophic microbial

activity. In addition to providing a template for a diverse set of ecological interactions, the

marine macrobiota studied here hosted a diverse set of microbial metabolisms and enhanced

rates of carbon and nitrogen cycling in coastal ecosystems.

*Code and data availability.* Available from the authors upon request.

*Competing interests.* The authors declare that they have no competing interests.

*Acknowledgements.* We are grateful to the Makah Tribal Council for access to Tatoosh Island

and Second Beach. We thank S. Betcher, L. Harris, G. Siegmund, K. Thomas, A.M., J. B. and J.

T. Wootton for help in the field, J. Larkum for analyzing seawater isotopes, and S. Pather for

foundational work. B. Weigel provided helpful feedback and discussion. The University of

Chicago Physics shop ably helped with the design and construction of the chambers. Funding

was provided by NSF-OCE 09-28232 (CAP), NSF-OCE 09-28152 (MAA). Captain R. McQuin

and the crew of the R/V C. Barnes, especially J. Postels, facilitated the offshore sampling.

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

## FIGURE CAPTIONS

Figure 1. The change in the concentrations of DIN (a. ammonium, b. nitrate, c. nitrite) in uM over the course of the experiment when only seawater was present, versus the addition of
bioballs or mussel or *Prionitis*. Nitrogen transformation rates (in nmol $L^{-1}$ $h^{-1}$) for d. ammonium oxidation, e. nitrate reduction, and f. the ammonium remineralization rate. Letters indicate statistical differences with ANOVA and Tukey HSD.

Figure 2. The ammonium oxidation rate (a.) and the nitrate reduction rate (b.) contrasted between day and night hours for mussels or seawater at the shore. Data are log-transformed
(from nmol $L^{-1}$ $h^{-1}$) to facilitate comparison. Rates with mussels were always greater (for a. $F_{1,11}$=52.59, p<0.001, and b. $F_{1,14}$=68.14, p<0.001). Ammonium oxidation rates in association with mussels or in seawater alone did not differ between day and night ($F_{1,11}$=0.58, p=0.461), while nitrate reduction in seawater was greater during the day ($F_{1,14}$=5.83, p=0.030).  Changes to the isotopes of inorganic nitrogen are shown in Figure A1.

Figure 3. a. The ammonium oxidation rate in surface seawater collected at the shore versus offshore 2-5 km, based on 4 trials in each locale in Jun and Jul of 2012.  The rates did not differ



(t= $t_{3.8}$=1.65, p=0.177). The change in DIN and Silica also did not differ between as a function of whether seawater was from the shore or offshore (b. DIN $t_{3.8}$=1.31, p=.260, c. Silica $t_{3.8}$=0.68, p=0.525). All rates in nmol $L^{-1}$ $h^{-1}$.

Figure 4. The effect of supplemental DOC on a. the rate of ammonium oxidation (in nmol $L^{-1}$ $h^{-1}$), b-g the change in nutrient concentrations (uM), and h. the oxygen concentration (in mg $L^{-1}$). An * indicates a significant difference (p<0.05) between the control and the DOC addition for each of seawater alone, or seawater with bioballs or the red alga *Prionitis*. ‡ indicates 0.10>p<0.05.







695        Figure 1

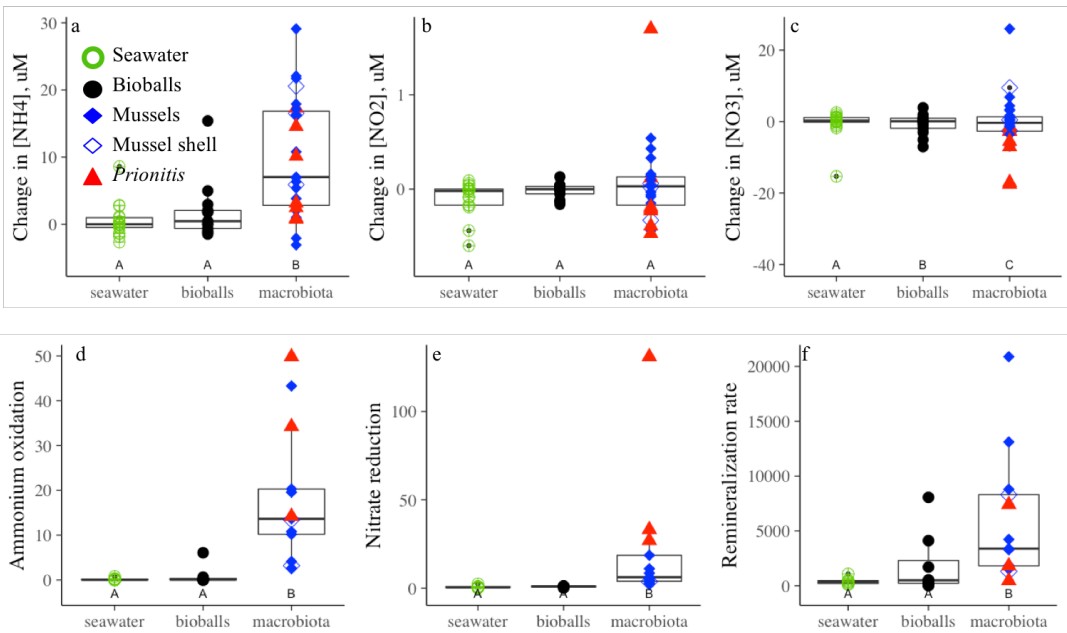



Figure 2

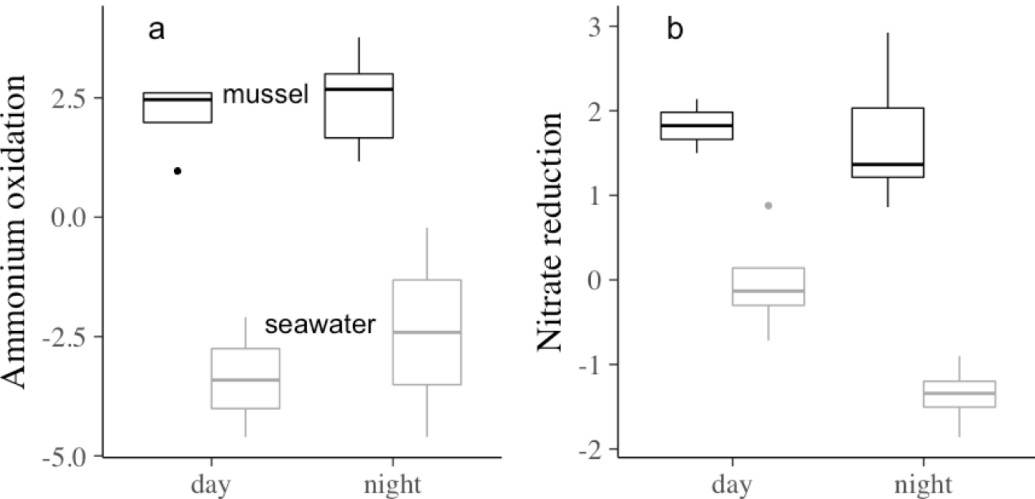




Figure 3

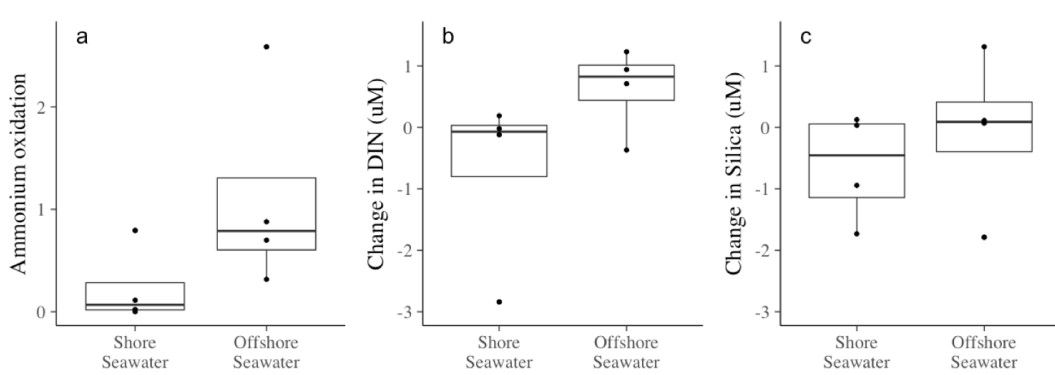



Figure 4

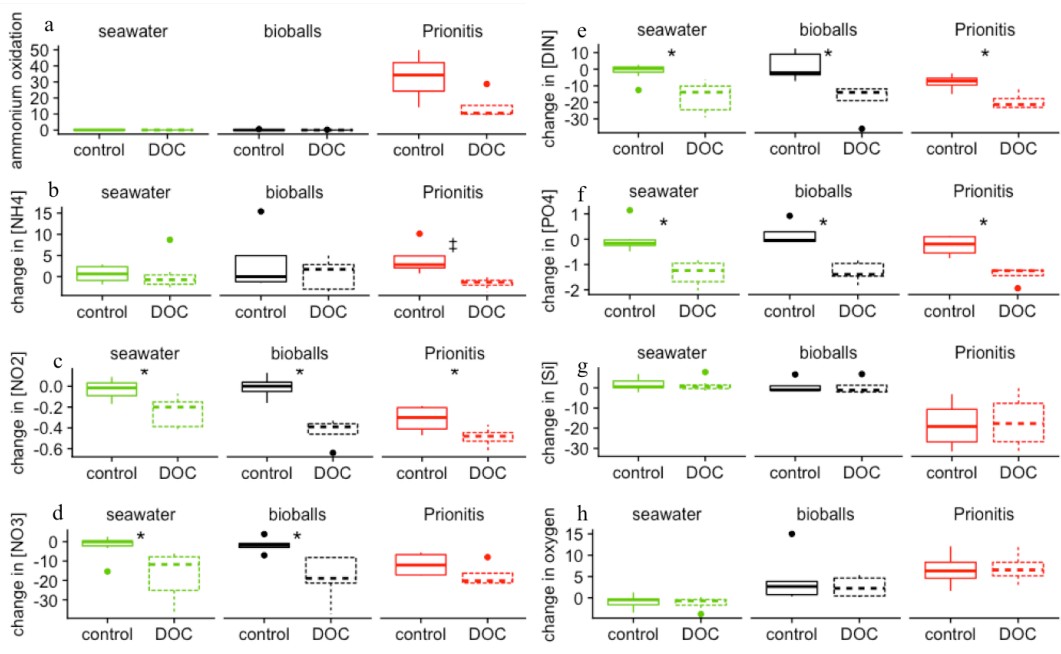

710                                            APPENDIX

Figure A1. An example of each substrate type in an enrichment with $^{15}N$-$NH_4^+$ (a, b) and
$^{15}N$-$NO_3^-$ (c, d). When $^{15}N$-$NH_4^+$ was added, enrichment in $\delta^{15}N_{NO2}$ (solid line) and $\delta^{15}N_{NO3}$
(dashed line) was measured (b), while enrichment in $\delta^{15}N_{NH4}$ (dotted line, c.) and $\delta^{15}N_{NO2}$ (d)
followed the addition of $^{15}N$-$NO_3^-$. Ammonium regeneration in the chambers, particularly
mussels, diluted the $\delta^{15}N_{NH4}$ signal (a. below and Fig 1f). Deviations in our target of initial
enrichment (10000 and 2000‰) occurred due to natural variation in nutrient concentrations at
the time of tracer addition.

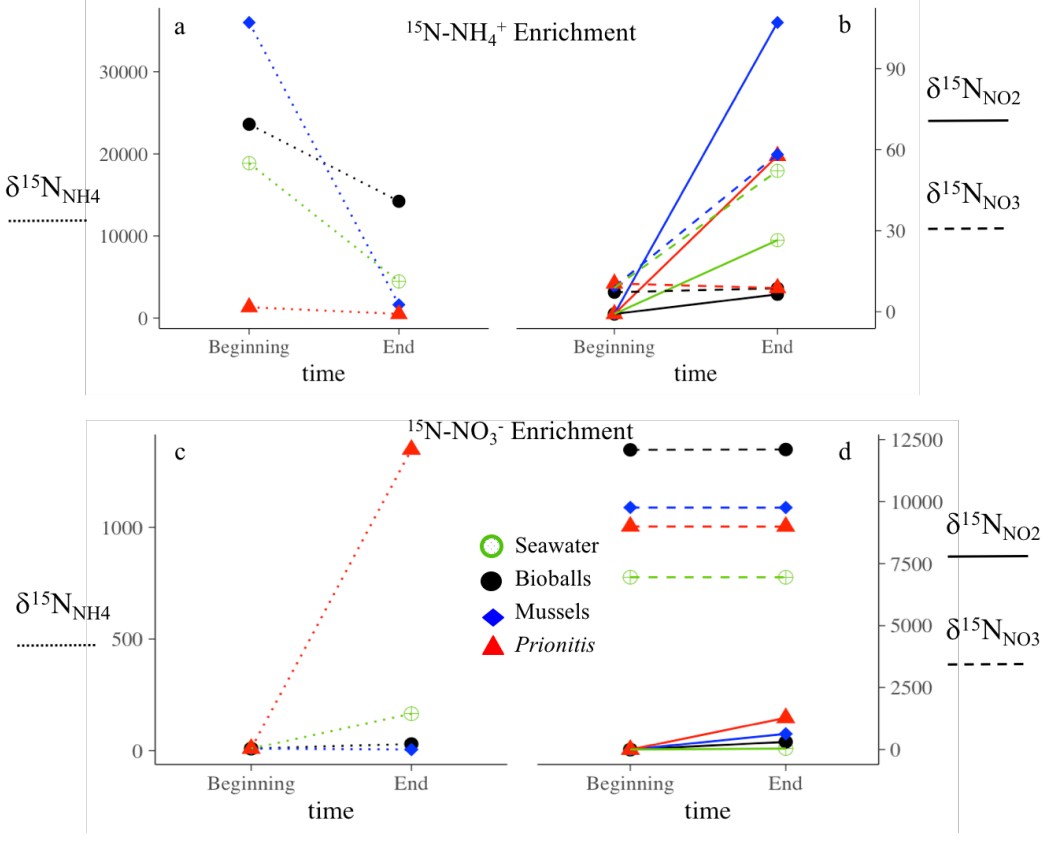