# Peer review of "Enhanced microbial nitrogen transformations in association with macrobiota from the rocky intertidal"

_Biogeosciences, 2018_

## Referee Comment (RC1) · Anonymous Referee #1 · 27 Jun 2018

General comments:

This article seeks to tease apart the effects of coastal biota and the settlement surface they provide on microbial nitrogen cycling. The article further aims to determine how this is influenced by factors including light vs dark and addition of glucose (mimicking provision of DOC via excretion from biota).

Understanding of the role of macrobiota communities in coastal nitrogen cycling sits well within the scope of the Biogeosciences journal and is particularly important given the widespread increase in coastal nitrogen concentrations and interest in the ability of coastal habitats to cope with or buffer against this.

The paper is not overly novel, but provides new data for this area of research and demonstrates the potential for changes in macrobiota to alter coastal N processing, which is of general interest. The authors could broaden the scope/interest of the paper by providing comment on how applicable these results are likely to be for other biota and regions, and by providing additional background information on environmental conditions, etc. It would also be helpful to see some further discussion of the possible mechanisms underlying the role of biota (vs inert substrate) in eliciting changes in microbial N transformations.

Overall, the manuscript would benefit from re-working to improve clarity, particularly relating to the methods and statistical analyses used. I found this section to be confusing, making it difficult to ascertain how reliable/robust the results are.

Please see the supplement for specific comments and technical corrections.

Please also note the supplement to this comment:
https://www.biogeosciences-discuss.net/bg-2018-198/bg-2018-198-RC1-supplement.pdf

**Supplement:**

**General comments:**

This article seeks to tease apart the effects of coastal biota and the settlement surface they provide on microbial nitrogen cycling. The article further aims to determine how this is influenced by factors including light vs dark and addition of glucose (mimicking provision of DOC via excretion from biota).

Understanding of the role of macrobiota communities in coastal nitrogen cycling sits well within the scope of the Biogeosciences journal and is particularly important given the widespread increase in coastal nitrogen concentrations and interest in the ability of coastal habitats to cope with or buffer against this.

The paper is not overly novel, but provides new data for this area of research and demonstrates the potential for changes in macrobiota to alter coastal N processing, which is of general interest. The authors could broaden the scope/interest of the paper by providing comment on how applicable these results are likely to be for other biota and regions, and by providing additional background information on environmental conditions, etc. It would also be helpful to see some further discussion of the possible mechanisms underlying the role of biota (vs inert substrate) in eliciting changes in microbial N transformations.

Overall, the manuscript would benefit from re-working to improve clarity, particularly relating to the methods and statistical analyses used. I found this section to be confusing, making it difficult to ascertain how reliable/robust the results are.

**Specific comments**

Title: The title could be improved. This seems to be the only place in the manuscript where it is specified that the biota are intertidal. Furthermore, the incubations were (apparently) fully submerged.

Abstract: The abstract seems to be missing some key information, and there appear to be some inconsistencies with the body of the manuscript, as follows:

- Why is a day/night comparison only described for mussels? Was this comparison done for other substrates? This is not specified in the methods section of the manuscript.
- DOC added to which treatments/substrates? From the abstract it sounds like DOC was added only to chambers with mussels, but then in the methods section is seems like DOC was added to algae, seawater and bioballs, but not mussels.
- No mention of mussel shells (mentioned elsewhere). Is this considered 'inert substrate', in which case the abstract should probably specify the various inert substrates that were used.
- Offshore is specified as 2-5km, whereas ln 112 specifies 1-5km and ln 127 specifies 2-3km.
- Abstract states day and night rates were similar, but results state that nitrate reduction was higher in the day (pg 13).

Introduction: The introduction seems lengthy and could be more concise. For example, there is much mention of N release by biota and how this might affect N cycling, but this is not tested, so could be considerably shortened or removed. However, it should be made clear why the excreted C was considered to be potentially more important than excreted N.

Pg 3: There are papers that compare different habitats characterised by different biota that also provide some insight into the potential role of macrobiota. These seem to have been overlooked. For example, B. Eyre has a number of papers comparing biogeochemistry of different benthic

habitats (with and without seagrass, with and without burrowing animals) that could provide additional context (e.g. DOI 10.1007/s10533-010-9425-6).

Ln 108: Here it states that 15N-enrichment allowed for estimation of NH4 and NO3 fluxes, but this can be measured without 15N enrichment. The reason for the 15N enrichment is not made clear. It should be clearly stated how this following transfer of 15N from NOx/NH4 to NH4/NOx allows identification and quantification of the pathways responsible for N fluxes.

It should be clearly stated why nitrification and nitrate reduction are important processes.

Methods:

- Two chambers were used, so incubations were run in pairs and clearly over some period. How were these paired up to avoid the possibility that temporal variations in light/temperature (for example) may have affected apparent differences between the different treatments?
- It is not immediately clear, 54 assays of 2 chambers = 54 total incubations (26 pairs) or 108 incubations (54 pairs).
- Were the chambers completely submerged within the tide pools? What was the depth of overlying water?
- What were the 'natural light and temperature conditions'? How much did they vary within an incubation and among incubations?
- What mass of substrate (mussels, algae, bioballs, etc.) were used within each chamber?
- Ln 141: Why was the cooler shaded? What depth was the water bath. What temperature was the water bath held at? What were the light conditions?
- Why was red algae used at the Tatoosh Island site and not at Second Beach. Indeed, given that the incubations were run for substrates ex situ, why were multiple sites necessary? Were the mussels and algae collected locally to each site?
- There is no mention of mussel shell incubations, yet data for these are shown in the results (e.g. fig 1).
- For ship-based incubations, what were the incubation conditions? Also, it is not clear how many incubations were done, and what was compared. It is not at all clear to me what is meant by "4 replicates of each shore and offshore chambers", particularly with n=9 (ln 128).
- Ln 126: 'added an enrichment of 10000‰ of d15NH4'. I suggest rewording, as this does not really make sense as written. Suggest '0.05M 15N-labeled NH4Cl was added to give a final approximate d15N value of 10000‰. Similarly, reword statement for addition of 15N-NO3.
- Ln 159: 'dissolved' N2 gas.
- Ln 162: How was the chamber agitated?
- Ln 165: How were samples stored? With/without headspace? What temperature?
- Section 2.2: What treatments had DOC added? None with mussels?
- Ln 175: Clarify – 4 paired runs each for Prionitis and bioballs?
- Ln 176: First mention of 15N-enrichment here comes 'out of the blue'. Why NH4 and not NO3 for the enrichment in this part of the study?
- There appears to be no mention of the day/night comparison
- There is no description of the statistical analyses used and referred to in the results.

Results:

- Ln 236: Mentions chambers during daylight hours, but what about outside of daylight hours? Are only daylight incubations shown on Fig 1?

- Ln 239-240: Nitrate and nitrite are swapped in the text or in Fig 1. Check for any impact on other sections of the text.
- Ln 241-243: Make clear that this describes changes in d15N following 15N addition. However, this section also seems to be repeat of the methods. No results are presented here.
- Ln 251: Is this an average across all bioball treatments?
- Ln 256: First mention of mussel shells. Discussion material here.
- Ln 260: 'than that for'
- Ln 257-260: Rewrite. Remineralization twice.
- Ln 278: Interaction term is not significant at p=0.05, in which case nitrate reduction was also lower for mussels at night.
- Ln 294: No mention in methods of DOC measurements. What about mussels? Did they increase DOC concentration too?
- Pg 14: Some discussion material in the second paragraph.

Discussion

The discussion is thorough in the most part and makes good use of previous work to support the conclusions made. However, there are a few parts of the results I would have liked to see further explained: 1) If the microbial communities associated with biota have such an effect on N processing, yet excreted DOC from biota apparently has little effect, what alternative mechanism might explain the fact that the inert substrates (also with microbial communities) had little effect on N processing? Are the communities support by inert substrates (including mussel shells) and macrobiota different? 2) DOC is made up of a variety of compounds. Glucose is one of the very labile compounds and would presumably be most likely to elicit a response from the microbial community. Is this why this specific compound was chosen? Is this representative of the DOC that would be assumed to be provided by the macrobiota? 3) So macrobiota change N processing. What is the significance of these changes? Why is this important?

I also had a few more minor notes:

- Is Prionitis common and/or a typical macroalga? Is it expected to affect N processing in a similar way to other macroalga?
- Ln 329: It is not clear exactly what data was used from Paine 2002 in calculations – just the mass estimates?
- Ln 423: Was N2 not detected because denitrification did not occur, or because the rate was too low to be detected (or not enough label was applied)? This could possibly be determined by calculating the lowest N2 flux that could have realistically been detected based on the amount of 15N added (and allowing for instrument error, etc.).

**Technical corrections**

- Ln 16-17: Sentence needs re-writing. Role for microbial activity? Or inert substrate for microbial activity? Clarify please.
- Ln 17: 'only seawater' as a control (all the chambers had seawater).
- Ln 19: Change 'of seawater' to 'in seawater'
- Ln 19-20: Remove "effect of simply an"
- Ln 28: Change 'elevating the concentration of DOC' to 'DOC addition'
- Ln 28 and 30: 'indicate' used twice close together. Find an alternative?
- Ln 54: 'enhance' should be 'enhances'

- Ln 60: manipulated DOC 'concentration'
- Ln 74: Remove full stop after 'animals'
- Ln 81: Full stop missing from 'e.g.'
- Pg 5 (check for same elsewhere): $NH_4$ should be $NH_4^+$
- Ln 101: "The effects macrobiota have on both nitrogen excretion and DOC release are poorly understood." But this was not tested?
- Ln 112: Reword – e.g., does the microbial activity differ for seawater collected from nearshore and offshore. Also, not inconsistency in distance offshore vs other locations in the text.
- Ln 114: Remove 'experimental'.
- Ln 120: 'both retain and lose' replace with 'contribute to loss and retention'
- Ln 126: Remove extra 'W' from co-ordinates.
- Ln 127: 2-3km, vs 2-5 or 1-5 elsewhere
- Ln 130: Remove extra bracket before 'Pfister'
- Ln 160: 'is' should be 'are'. Space missing in umol/L
- Ln 165: injected 'this'
- Ln 214: Reword to 'estimated ammonium oxidation by monitoring 15N enrichment in nitrate following 15NH4 addition. Similarly for nitrate reduction sentence that follows.
- Ln 225: model 'to the' decline…
- Ln 230: mean concentration of ammonium = mean of what? Start and end? A number of replicates?
- Ln 274: Figure reference.
- Ln 278: Full stop missing from e.g.
- Switching between 'ammonia oxidation' and 'nitrification' could be confusing.
- Ln 288: Remove 'either'
- Ln 293: Space missing before 1000
- Ln 321: replace 'on' with 'for'
- Ln 329: mm should be mmol
- Ln 331: what about nitrate reduction?
- Ln 335: remove 'via'
- Ln 338: change to 'metabolism to offshore seawater'
- Ln 345: Any chance nearshore enrichment could reflect wastewater input?
- Ln 368: Change to 'glucose additions have resulted in decreased nitrification' (delete 'with DOC')
- Ln 369: decreased oxygen 'concentration'
- Ln 374: comma after microbes
- Ln 378: 'explanation to explain' Reword.
- Ln 391: remove comma before Joo
- Ln 655: 'mussel' should be 'mussels'
- Ln 666: 2-5km offshore
- Ln 667: Refer to parts b and c of the figure upon first mention of DIN and silica. Re-write this sentence (doesn't make sense as currently written).
- Ln 711: Replace 'in an' with 'following
- Figure 1: 1) Mussel shells come as a surprise. No mention of these incubations in the methods. 2) Why is data for Prionitis and mussels shown together and not in separate bars? 3) Difficult to distinguish the two sets of data in this format. 4) Show units for d, e, f on the figure (and on all other figures in the manuscript). 5) What are the lines connecting some data points? Error bars..?
- When referring to enriched ammonium and nitrate, better to say "15N-enriched" to avoid confusion, as many papers use the term enriched to refer to higher concentration (which you also have in here, albeit for DOC).

- Numerous places where ref formatting has problems such as extra brackets, first names included, names within brackets that should be part of the text, etc. (e.g. ln 37, 51, 81, 201, 328, 369, 380, 397)
- Check for instances where u should be μ. E.g. ln 172, 287, 292, 294, 654, Figure 1 axis labels.
- Check reference list for capitalization within some refs and not others: e.g. Azam, Capone, Croll, Diner, de Goeij, McIlvin, Offre, Paine, Worm, Zehr, Zhang.
- Reference Beman (ln 470) has first names in full.
- Ln 481: Space missing 'ina'
- Ln 466: Mytilus
- Ln 477: subscript
- Italics for scientific names in references Flombaum, Joo, Jacobs, Bayne

---

## Referee Comment (RC2) · Anonymous Referee #2 · 14 Jul 2018

The manuscript addresses roles of microbes associated with mussels, Mytilus californianus, and macro algae, Prionitis sternbergii, in nitrogen processing in coastal environment by experimental approach using enclosed chambers. The approach used in this study and presented results are not novel. Furthermore it is difficult to apply the knowledge obtained from a particular experimental condition to other environments because of lack of description of environmental and experimental condition. Although this topic fits well to the scope of the journal BGD, authors need to improve the discussion experimental condition and limitation of application to other environments. Further, the data presentation is inappropriate (see General comments). Thus the manuscript need major revision before publishing Biogeoscinece.

General comments: A) Was biomass of animals or macroalgae uniform in each chamber? I guess that they were not uniform because authors presented rates in per gram in the different paragraph. Units of Y-axis in Figs 1, 2 and 4 are not in per gram. Does the mean values of rates or differences in concentration obtained from chamber containing different biomass make sense? Whether does the variations in the figures depend on the difference in metabolic rate per unit biomass among individuals or in total biomass? Further, it is hard to understand aim of box in Fig 1 combining data obtained from different experimental conditions. How readers compare the rates and the differences obtained bioballs with those of macrobiota? In biomass of microbes dwelling on the surface or surface area? Similarly, the rates and the differences obtained mussel shell should be compared with those of living mussel after normalization with the surface area or with dry mass of shell. Because the data treatment could influence following statistical analysis comparing mean values, authors clarify/improve the data presentation. I cannot decide whether conclusion is based on appropriate analysis or not in the present style.

B) Addition of glucose might be one of the extreme case of DOC enrichment. C/N ratio of bacterial biomass and organic substrate affect uptake/release of DIN by heterotrophic bacteria (Kirchamn 2012 Processes in microbial ecology. Oxford University Press). Why authors choose glucose, which does not contain nitrogen? Is glucose major component of macroalgal exudate? I feel mismatch DOC between the term DOC used thorough the manuscript and glucose although glucose is DOC. Authors should clarify the aim of experimental addition DOC in the last part of introduction or discuss the difference between algal exudate reported in literature and glucose.

Specific comments: 1) Abstract Lines 26-27: "When we experimentally added DOC (glucose) as a carbon source, there was no change to nitrification rates." As described in general comments, please discuss rates in per gram of biomass.

2) Introduction: I feel that this section could be shortened.

3) Lines 38-42: This sentence is too long. Please separate the sentence.

4) Lines 83-85: Add reference.

5) Lines 89-90: Add reference.

6) Lines 130-132: Because incubation period did not cover day time, time should be described. Since photosynthesis affects nitrogen cycling in the chamber as authors state, light condition should be described as if authors compare the results of experiments conducted at the same time. And depth level at which the chambers were put should be added.

7) Lines 134-135: Biomass of macrobiota and mass/area of mussel sell should be added.

8) Lines 135-139: Add total amount or total surface area put into the chamber.

9) Lines 151-152: Did incubation conducted at in situ temperature?

10) Lines 159-161: Is there reference showing that nitrogen metabolism of microbes are saturated in these concentration?

11) Lines 176-178: What is final concentration of NH4+ in the chamber after addition of 15NH4Cl?

12) Lines 217-219: The flux in inorganic nitrogen from where to where? Please clarify them.

13) In Materials and Methods section: Statistical analysis should be explained.

14) In result section: all of rates and difference should be presented in per gram and with SD or SE. In my opinion the results should be presented with its variation. Thus SD is appropriate rather than SE.

15) Line 269: Which graph should be referred to compare with data at night? In the next section? Please rearrange the paragraphs.

16) Section 3.4: Description of biomass of microbes and DIN concentration should be presented. The analysis is poor in the present style.

17) Lines 294-295: the DOC concentration? 9.3 mmol C L-1 per hour?

18) Lines 306-307: Assimilation rather than respiration with glucose enrichment.

19) Lines 321-323: Could amounts of substrates (ammonium and nitrate) support these potential consumption rate? The determined rates were obtained in enriched experimental condition in the chamber.

20) Line 393: the C:N of what?

21) Figures 1, 2 and 4: Values should be presented in per gram or area.

22) Figure 1: Each group of macrobiota should be presented separately.

END OF REVIEW

---

## Author Comment (AC1) · 29 Jul 2018

To the Editor: 28 Jul 2018

We were grateful to receive such thorough reviews of our ms bg-2018-198, "*Enhanced microbial nitrogen transformations in association with macrobiota from the rocky intertidal.*" We have responded to all comments below using italicized font for our reply.

Sincerely,
Catherine Pfister & Mark Altabet

General comments:
This article seeks to tease apart the effects of coastal biota and the settlement surface they provide on microbial nitrogen cycling. The article further aims to determine how this is influenced by factors including light vs dark and addition of glucose (mimicking provision of DOC via excretion from biota). Understanding of the role of macrobiota communities in coastal nitrogen cycling sits well within the scope of the Biogeosciences journal and is particularly Important given the widespread increase in coastal nitrogen concentrations and interest in the ability of coastal habitats to cope with or buffer against this.
The paper is not overly novel, but provides new data for this area of research and demonstrates the potential for changes in macrobiota to alter coastal N processing, which is of general interest. The authors could broaden the scope/interest of the paper by providing comment on how applicable these results are likely to before other biota and regions, and by providing additional background information on environmental conditions, etc. It would also be helpful to see some further discussion of the possible mechanisms underlying the role of biota (vs inert substrate) in eliciting changes in microbial N transformations.
Overall, the manuscript would benefit from reworking to improve clarity, particularly relating to the methods and statistical analyses used. I found this section to be confusing, making it difficult to ascertain how reliable/robust the results are.
*We thank the reviewer for the suggestions and editorial corrections. There were many careful edits suggested that have improved our presentation of the research. We have expanded the breadth of the presentation and added more detail on the environmental parameters related to the experiments.*
Specific comments
Title:
The title could be improved. This seems to be the only place in the manuscript where it is specified that the biota are intertidal. Furthermore, the incubations were (apparently) fully submerged.
*Title was changed.*
Abstract:
The abstract seems to be missing some key information, and there appear to be some inconsistencies with the body of the manuscript, as follows:
Why is a day/night comparison only described for mussels? Was this comparison done for other substrates? This is not specified in the methods section of the manuscript.
*Now specified*
DOC added to which treatments/substrates? From the abstract it sounds like DOC was added only to chambers with mussels, but then in the methods section is seems like DOC was added to algae, seawater and bioballs, but not mussels.

*Now specified*

No mention of mussel shells (mentioned elsewhere). Is this considered 'inert substrate', in which case the abstract should probably specify the various inert substrates that were used.
*We now corrected this omission – here and in the Methods*

Offshore is specified as 2-5km, whereas ln 112 specifies 1-5km and ln 127 specifies 2-3km.
*It was 2-5 km. Corrected throughout.*

Abstract states day and night rates were similar, but results state that nitrate reduction was higher in the day (pg 13). *Corrected*

Introduction:

The introduction seems lengthy and could be more concise. For example, there is much mention of N release by biota and how this might affect N cycling, but this is not tested, so could be considerably shortened or removed.
*We omitted some text about animal excretion while trying to add some clarity about the novelty in the Introduction.*

However, it should be made clear why the excreted C was considered to be potentially more important than excreted N.
*Done. We saw this as relevant for the alga Prionitis because algae excrete DOM (Line 80).*

Pg 3:

There are papers that compare different habitats characterised by different biota that also provide some insight into the potential role of macrobiota. These seem to have been overlooked. For example, B. Eyre has a number of papers comparing biogeochemistry of different benthic habitats (with and without seagrass, with and without burrowing animals) that could provide additional context (e.g. DOI 10.1007/s1053301094256).
*We thank the reviewer and did not intend to overlook a lot of important studies in burrowing animals and soft sedimentary environments. We acknowledge that there are many instances where biogeochemical cycling and microbial activity are studied in soft sediments. There has been far fewer instance in rocky shore areas. The second paragraph of the introduction now reflects this.*

Ln 108:

Here it states that 15N enrichment allowed for estimation of NH4 and NO3 fluxes, but this can be measured without 15N enrichment. The reason for the 15N enrichment is not made clear. It should be clearly stated how this following transfer of 15N from NOx/NH4 to NH4/NOx allows identification and quantification of the pathways responsible for N fluxes.
*We added some explanatory text, but left details to Methods starting Line 166 and section 2.3.*

It should be clearly stated why nitrification and nitrate reduction are important processes.
*Done. Last paragraph of introduction.*

Methods:

Two chambers were used, so incubations were run in pairs and clearly over some period. How were these paired up to avoid the possibility that temporal variations in light/temperature (for example) may have affected apparent differences between the different treatments?

*The need to run replicates through time was a concern. By using paired contrasts, however, we were able to minimize the effects of any environmental variable on a particular treatment.*

It is not immediately clear, 54 assays of 2 chambers = 54 total incubations (26 pairs) or 108 incubations (54 pairs).
*We had 27 pairs and we now state this, Line 124.*
-
Were the chambers completely submerged within the tide pools? What was the depth of overlying water?
*The water was typically 15 cm or about halfway up the chamber. Now stated.*

What were the 'natural light and temperature conditions'? How much did they vary within an incubation and among incubations? *Description added, starting L 134.*
*We had extensive temperature data, but light was a highly variable parameter and was not measured continuously. Because seawater is generally colder than air temperature at this time of the year, the temperature tended to track any increases in light level.*
-
What mass of substrate (mussels, algae, bioballs, etc.) were used within each chamber?
*Measurements added, L 137 to L 156.*

Ln 141: Why was the cooler shaded? What depth was the water bath. What temperature was the water bath held at? What were the light conditions?
*The cooler was shaded to prevent artificially large excursions in water temperature. See comments above on temperature and light and L135 in revised ms.*
-
Why was red algae used at the Tatoosh Island site and not at Second Beach. Indeed, given that the incubations were run for substrates ex situ, why were multiple sites necessary? Were the mussels and algae collected locally to each site?
*Mussels and algae were collected locally to each site. Multiple sites were used to increase the range of what could be done – within tidepools at Second Beach, and nearly continuously at the shore at Tatoosh Island, using seawater and organisms at the source. Explanations L 156.*
-
There is no mention of mussel shell incubations, yet data for these are shown in the results (e.g. fig 1).
*Corrected, L 155*
-
For ship based incubations, what were the incubation conditions? Also, it is not clear how many incubations were done, and what was compared. It is not at all clear to me what is meant by "4 replicates of each shore and offshore chambers", particularly with n=9 (ln 128).
*More explanation now given, starting L 166*
-
Ln 126: 'added an enrichment of 10000‰ of d15NH4'
*Done*

I suggest rewording, as this does not really make sense as written. Suggest '0.05M 15Nlabeled NH4Cl was added to give a final approximate d15N value of 10000‰. Similarly, reword statement for addition of 15N-NO3.
*Section reworded.*
-
Ln 159: 'dissolved' N2 gas.
*Done*
-
Ln 162: How was the chamber agitated?
*by simply hand mixing every hour. Details now added at L180*
-
Ln 165: How were samples stored? With/without headspace? What temperature?
*Details now added at L182*

Section 2.2: What treatments had DOC added? None with mussels?
*Correct*
-
Ln 175: Clarify –4 paired runs each for Prionitis and bioballs?
*More detail added to Section 2.2*
-
Ln 176: First mention of 15N enrichment here comes 'out of the blue'. Why NH4 and not NO3 for the enrichment in this part of the study?
*More detail was added at the beginning of section 2.2. We were specifically testing if heterotrophic nitrification increased.*
-
There appears to be no mention of the day/night comparison
*Details now at L159.*
-
There is no description of the statistical analyses used and referred to in the results.
*Added at end of Methods 2.3.*

Results
Ln 236: Mentions chambers during daylight hours, but what about outside of daylight hours? Are only daylight incubations shown on Fig 1?
*Clarified, 1^st sentence of Results.*
-
Ln 239-240: Nitrate and nitrite are swapped in the text or in Fig 1. Check for any impact on other sections of the text.
*Oops, fixed.*
-
Ln 241-243: Make clear that this describes changes in d15N following 15N addition. However, this section also seems to be repeat of the methods. No results are presented here.
*These are results that illustrate the transfer of the isotope. They were added as a supplement to help the reader see raw isotope dynamics in addition to the analysis of rates.*
-
Ln 251: Is this an average across all bioball treatments?

*Yes, now stated.*

-

Ln 256: First mention of mussel shells. Discussion material here.
*Now introduced in Abstract, Methods*

-

Ln 260: 'than that for' *corrected*

-

Ln 257-260: Rewrite. Remineralization twice.
*corrected*

-

Ln 278: Interaction term is not significant at p=0.05, in which case nitrate reduction was also lower for mussels at night.
*Edited this to reflect this statement, L302*

-

Ln 294: No mention in methods of DOC measurements. What about mussels? Did they increase DOC concentration too?
*Put DOC measurements in Methods, section 2.3. Again, the DOC addition experiment was not done with mussels.*

-

Pg 14: Some discussion material in the second paragraph.
*The inference we were attempting to make with glucose dynamics seemed to necessitate leading the reader through these results. They seem more logically here than in Discussion and so we left these few sentences.*

Discussion
The discussion is thorough in the most part and makes good use of previous work to support the conclusions made. However, there are a few parts of the results I would have liked to see further explained: 1) If the microbial communities associated with biota have such an effect on N processing, yet excreted DOC from biota apparently has little effect, what alternative might explain the fact that the inert substrates (also with microbial communities) had little effect on N processing?Are the communities support by inert substrates (including mussel shells) and macrobiota different?
*Yes. Previous genomic work by Pfister et al. 2014 is cited L 369, L420 and we generally tried to better highlight that we have verified that microbial taxa have some distinctions on these species and substrates.*

2) DOC is made up of a variety of compounds. Glucose is one of the very labile compounds and would presumably be most likely to elicit a response from the microbial community.
Is this why this specific compound was chosen? Is this representative of the DOC that would be assumed to be provided by the macrobiota?
*We added rationale to the Methods, Section 2.2. Glucose is common to many species and used in other studies. We recognize, however, that there are many other potentially important and relevant compounds and we hope to incorporate this in future research.*

3) So macrobiota change N processing. What is the significance of these changes? Why is this important?

*This is an important point and we expanded upon this in the Discussion, Section 4.1*

I also had a few more minor notes:
Is Prionitis common and/or a typical macroalga? Is it expected to affect N processing in a similar way to other macroalga?
*We introduce the species on L100. We do not know whether Prionitis is unique or typical and won't until we have information from other species. However, there are an increasing number of studies that demonstrate microbial communities on macroalgae.*
-
Ln 329:
It is not clear exactly what data was used from Paine 2002 in calculations just the mass estimates?
*Yes. We reworded this part for clarity.*
-
Ln 423: Was N2 not detected because denitrification did not occur, or because the rate was too low to be detected (or not enough label was applied)? This could possibly be determined by calculating the lowest N2 flux that could have realistically been detected based on the amount of 15N added (and allowing for instrument error, etc.).
*This is now better addressed on section 4.3.*

Technical corrections
Ln 16-17: Sentence needs rewriting. Role for microbial activity? Or inert substrate for microbial activity? Clarify please. *Reworded*
-
Ln 17: 'only seawater' as a control (all the chambers had seawater). *Done*
-
Ln 19: Change 'of seawater' to 'in seawater' *Done*
-
Ln 19-20: Remove "effect of simply an" *Done*
-
Ln 28: Change 'elevating the concentration of DOC' to 'DOC addition' *Done*
-
Ln 28 and 30: 'indicate' used twice close together. Find an alternative? *Changed*
-
Ln 54: 'enhance' should be 'enhances' *Done*
-
Ln 60: manipulated DOC 'concentration' *Done*
-
Ln 74: Remove full stop after 'animals' *Done*
-
Ln 81: Full stop missing from 'e.g.' *Done*
-
Pg 5 (check for same elsewhere): NH4 should be NH4+ *Corrected throughout*
-
Ln 101: "The effects macrobiota have on both nitrogen excretion and DOC release are poorly understood." But this was not tested? *Reworded starting L95*

-
Ln 112: Reword e.g., does the microbial activity differ for seawater collected from nearshore and offshore. Also, not inconsistency in distance offshore vs other locations in the text. *Done*
-
Ln 114: Remove 'experimental'. *Done*
-
Ln 120: 'both retain and lose' replace with 'contribute to loss and retention' *Done*
-
Ln 126: Remove extra 'W' from coordinates. *Done*
-
Ln 127: 2-3km, vs 2-5 or 1-5 elsewhere  *Done, 2-5 km*
-
Ln 130: Remove extra bracket before 'Pfister' *Done*
-
Ln 160: 'is' should be 'are'. Space missing in umol/L *Done*
-
Ln 165: injected 'this' *Done*
-
Ln 214: Reword to 'estimated ammonium oxidation by monitoring 15N enrichment in nitrate following 15NH4 addition. Similarly for nitrate reduction sentence that follows. *Reworded*
-
Ln 225: model 'to the' decline... *Done*
-
Ln 230: mean concentration of ammonium= mean of what? Start and end? A number of replicates? *Start and end. Explained in text.*
-
Ln 274: Figure reference. *Done*
-
Ln 278: Full stop missing from e.g. *Done*
-
Switching between 'ammonia oxidation' and 'nitrification' could be confusing. *We thought it was worthwhile to use both*
-
Ln 288: Remove 'either' *Done*
-
Ln 293: Space missing before 1000 *Done*
-
Ln 321: replace 'on' with 'for' *Done*
-
Ln 329: mm should be mmol *Done*
-
Ln 331: what about nitrate reduction? *In reviewing the Reviewers comments, we discovered a mistake here and we revised this part, adding clarity about the nitrate reduction part. Briefly, CP missed a umol to mmol conversion and thus was overestimating the macroalgal contribution. We have revised the contribution, noting that algae are an order of magnitude less than the mussels. Revisions start Section 4.1*

-

Ln 335: remove 'via' *Done*

-

Ln 338: change to 'metabolism too ffshore seawater' *Done*

-

Ln 345: Any chance nearshore enrichment could reflect wastewater input? *Human wastewater affects are likely negligible here. The region has very low human population size relative to the land area and the shores are heavily forested.*

Ln 368: Change to 'glucose additions have resulted in decreased nitrification' (delete 'with DOC') *Done*

-

Ln 369: decreased oxygen 'concentration' *Done*

-

Ln 374: comma after microbes *Done*

-

Ln 378: 'explanation to explain' *Reworded.*

-

Ln 391: remove comma before Joo *Done*

-

Ln655: 'mussel' should be 'mussels' *Done*

-

Ln 666: 2-5km offshore *Done*

Ln 667: Refer to parts b and c of the figure upon first mention of DIN and silica. Re write this sentence (doesn't make sense as currently written). *Done*

-

Ln 711: Replace 'in an' with 'following *Done*

-

Figure 1:1) Mussel shells come as a surprise. No mention of these incubations in the methods. *Corrected as stated above.*

2) Why is data for Prionitis and mussels shown together and not in separate bars? *We were interested in emphasizing the role of macrobiota generally. However, we note that both reviewers wanted these data separated into a box for each species and so have done this.*

3) Difficult to distinguish the two sets of data in this format *We were interested in emphasizing the role of macrobiota generally, but note that both reviewers wanted these data separated into a box for each species and so have done this.*

4) Show units for d, e, f on the figure (and on all other figures in the manuscript). *We chose to leave the units in the fig caption because the figure gets crowded with the units and the font size has to be reduced to an extent that reading it is difficult.*

5)What are the lines connecting some data points? Error bars..? *We now explain the features of our boxplots in the Figure 1 caption.*

-

When referring to enriched ammonium and nitrate, better to say "15N-enriched" to avoid confusion, as many papers use the term enriched to refer to higher concentration (which you also have in here, albeit for DOC). *Done*

*Most of the edits below are all related to the use of Zotero. We have tried to correct all mistakes within the References.*
- Numerous places where ref formatting has problems such as extra brackets, first names included, names within brackets that should be part of the text, etc. (e.g. ln 37, 51, 81, 201, 328, 369, 380, 397)
- Check for instances where u should be μ. E.g. ln 172, 287, 292, 294, 654, Figure 1 axis labels. *Done, except for axis labels.*
- Check reference list for capitalization within some refs and not others: e.g. Azam, Capone, Croll, Diner, de Goeij, McIlvin, Offre, Paine, Worm, Zehr, Zhang.
- Reference Beman (ln 470) has first names in full.
- Ln 481: Space missing 'ina'
- Ln 466: Mytilus
- Ln 477: subscript
- Italics for scientific names in references Flombaum, Joo, Jacobs, Bayne

Anonymous Referee #2

The manuscript addresses roles of microbes associated with mussels, Mytilus califor-nianus, and macro algae, Prionitis sternbergii, in nitrogen processing in coastal envi-ronment by experimental approach using enclosed chambers. The approach used in this study and presented results are not novel. Furthermore it is difficult to apply the knowledge obtained from a particular experimental condition to other environments be-cause of lack of description of environmental and experimental condition. Although this topic fits well to the scope of the journal BGD, authors need to improve the discussion experimental condition and limitation of application to other environments. Further, the data presentation is inappropriate (see General comments). Thus the manuscript need major revision before publishing Biogeoscinece.

*We welcome the opportunity to revise the manuscript with these comments in mind, including the description of the local environment. Indeed, we thought a strength of the work was the fact that many experiments were done in situ in tidepools, while others were done immediately adjacent to the site where the organisms were collected; all used seawater collected immediately on site.*

General comments: A) Was biomass of animals or macroalgae uniform in each cham-ber? I guess that they were not uniform because authors presented rates in per gram in the different paragraph.
*Biomass was not uniform among individual animals and algae and there was thus natural variability in mass and we do present rates per gram. We put in more information on the mass in the Methods.*

Units of Y-axis in Figs 1, 2 and 4 are not in per gram. Does the mean values of rates or differences in concentration obtained from chamber containing different biomass make sense? Whether does the variations in the figures depend on

the difference in metabolic rate per unit biomass among individuals or in total biomass? Further, it is hard to understand aim of box in Fig 1 combining data obtained from different experimental conditions. How readers compare the rates and the differences obtained bioballs with those of macrobiota? In biomass of microbes dwelling on the surface or surface area? Similarly, the rates and the differences obtained mussel shell should be compared with those of living mussel after normalization with the surface area or with dry mass of shell. Because the data treatment could influence following statistical analysis comparing mean values, authors clarify/improve the data presentation. I cannot decide whether conclusion is based on appropriate analysis or not in the present style.

*We agree with the reviewer that it is difficult to know whether mass, surface area, or the area of a particular surface is the best determinant of microbial activity. Area estimates can be particularly hard to quantify. Instead, we used a metric of the size of a species that we could scale-up based on data from the shore. Thus, biomass allowed us to scale up these rates to Paine's algal densities and Wootton's mussel densities. Because we were always comparing our addition of a seaweed, animal or inert substrate to seawater, it was appropriate to compare the rates in a volume of seawater. Indeed, it was our goal to investigate how species influence the processes in the water column. Thus, we show the rates in the figure as a volume measurement, but use the per mass estimates to scale up.*
*While Fig 1 showed all macrospecies together to illustrate the point that species have an effect, we appreciate that both reviewers want them separated and we have now done so in a revised Figure 1. As for comparing with bioballs, we agree that this is difficult comparison. However, bioballs are manufactured as microbial habitat for nitrogen transformations in commercial aquaria and yet their microbial activity is several orders of magnitude less than an individual alga.*

B) Addition of glucose might be one of the extreme case of DOC enrichment. C/N ratio of bacterial biomass and organic substrate affect uptake/release of DIN by heterotrophic bacteria (Kirchamn 2012 Processes in microbial ecology. Oxford University Press). Why authors choose glucose, which does not contain nitrogen? Is glucose major component of macroalgal exudate? I feel mismatch DOC between the term DOC used thorough the manuscript and glucose although glucose is DOC. Authors should clarify the aim of experimental addition DOC in the last part of introduction or discuss the difference between algal exudate reported in literature and glucose.

*As mentioned above to Reviewer 1, we added rationale to the Methods, Section 2.2. Glucose is common to many species and used in other studies (e.g. Zhang et al. 2013), so it allowed comparison. We recognize, however, that there are many other potentially important and relevant compounds (laminarin, mannitol, and more) and we hope to incorporate this in future research.*

Specific comments: 1) Abstract Lines 26-27: "When we experimentally added DOC (glucose) as a carbon source, there was no change to nitrification rates." As described in general comments, please discuss rates in per gram of biomass.

*We think it is important here to keep the units as per volume in our graphical presentation in order to compare to seawater. Seawater is well-studied with respect to microbial nitrogen processing and our point here is that macrospecies can be loci for these transformations and they will affect the surrounding seawater. We do use rates in terms of biomass when we scale up to rocky shores (Discussion). Also see below.*

2) Introduction: I feel that this section could be shortened.
*Specific editorial suggestions from Rev 1 led us to shorten this (see above).*

3) Lines 38-42: This sentence is too long. Please separate the sentence.
*Done.*
4) Lines 83-85: Add reference.
*Done.*

5) Lines 89-90: Add reference.
*Done.*

6) Lines 130-132: Because incubation period did not cover day time, time should be described. Since photosynthesis affects nitrogen cycling in the chamber as authors state, light condition should be described as if authors compare the results of experiments conducted at the same time. And depth level at which the chambers were put should be added.
*Chambers always had ~15 cm of water depth in coolers and tidepools. We did not, however, continuously monitor light levels. Daytime incubations always started by 6 am and ended by 6 pm, while nighttime incubations were under full darkness.*

7) Lines 134-135: Biomass of macrobiota and mass/area of mussel sell should be added.
*Biomass data is now more complete.*

8) Lines 135-139: Add total amount or total surface area put into the chamber.
*Inserted. Each ceramic ring has a surface area of approximately 6 cm2. We estimated a surface area for the bioballs between 15-20 cm2.*

9) Lines 151-152: Did incubation conducted at in situ temperature?
*Yes. More detailed added on temperature Line 168.*

10) Lines 159-161: Is there reference showing that nitrogen metabolism of microbes are saturated in these concentration?
*No. The nitrogen concentrations that determine microbial activity is unknown in these systems and we note that nitrate concentrations are high (at least 20 uM) due to upwelling and ammonium can reach 5 uM and more due to animal activity.*

11) Lines 176-178: What is final concentration of $NH_4^+$ in the chamber after addition of $^{15}NH_4Cl$?

*The concentration of NH4+ that was added as 15NH4Cl was trivial (<.01uM) compared to the amount already there (~1-2uM or more). Thus, the addition of 15NH4Cl did not markedly change concentration.*

12) Lines 217-219: The flux in inorganic nitrogen from where to where? Please clarify them.
*Oxidation and reduction. Clarified.*
13) In Materials and Methods section: Statistical analysis should be explained.

*More detail now put in at the end of Section 2.3. In general, the statistical analyses were ANOVA or paired t-tests and are stated in each part of the Methods.*

14) In result section: all of rates and difference should be presented in per gram and with SD or SE. In my opinion the results should be presented with its variation. Thus SD is appropriate rather than SE.

*Through our presentation of the data, we have shown the variation in all data and in all Figures using boxplots. We now provide a key to the boxplots in Figure 1 caption to make this clear. Specifically, we state that the box shows 50% of the data, the horizontal line is the median, and the vertical lines represent the first and fourth quartiles. Where vertical lines are absent, they are contained within the boxes. Outliers are shown as individual points and are 1.5 times the value beyond the top or bottom of the box. Note that the only place we use SE is in our estimates of per gram rates nitrogen metabolism for mussels and Prionitis in Section 3.2. We report the SE here because we then use these mean estimates to scale up and a measure of the dispersion around the mean was appropriate. Again, we report the measurements here in per g to scale up to previous censuses of mussels (Wootton) and algae(Paine) in this region. Otherwise, we report rates in per L to compare with seawater. Thus, rates in both metrics are available to the reader.*

15) Line 269: Which graph should be referred to compare with data at night? In the next section? Please rearrange the paragraphs.

*Agreed that this paragraph is out of place. We moved it to be the second paragraph in Section 3.3 and made clear that we are referencing Figure 2.*

16) Section 3.4: Description of biomass of microbes and DIN concentration should be presented. The analysis is poor in the present style.

*This paragraph has more detail to make clear the ammonium oxidation, and changes in DIN and Silica did not differ. We have no data on the biomass of microbes.*

17) Lines 294-295: the DOC concentration? 9.3 mmol C L-1 per hour?

*Clarified, per L.*

18) Lines 306-307: Assimilation rather than respiration with glucose enrichment.

*Or perhaps both? Reworded.*

19) Lines 321-323: Could amounts of substrates (ammonium and nitrate) support these potential consumption rate? The determined rates were obtained in enriched experimental condition in the chamber.

*It seems that it could. We did not increase ammonium or nitrate concentrations in the chambers; we used ambient levels. Further, the chamber has no flow or new nutrient supply as you would find in a natural coastal setting. Thus, it seems unlikely that we are overestimating the process.*

20) Line 393: the C:N of what?
*Clarified. Strauss and Lamberti termed this 'environmental' C:N, which presumes whatever substrate the microbial assemblages are interacting with.*

21) Figures 1, 2 and 4: Values should be presented in per gram or area.
*As mentioned above, we think it is important and logical to present the data as per Liter in the Figures that compare with seawater, but continue to use the per gram estimates for scaling up to the rocky shore.*

22) Figure 1: Each group of macrobiota should be presented separately.
*Done. Figure 1 is now revised to show each species.*

END OF REVIEW

---

## Author Response (AR2)

To the Editor:                                                                                  3 Nov 2018

We were grateful to have our revised ms bg-2018-198, "*Enhanced microbial nitrogen transformations in association with macrobiota from the rocky intertidal*" reviewed. We have responded to all comments below using italicized font for our reply. We thank the reviewers for their careful edits.

Sincerely,
Catherine Pfister & Mark Altabet

Anonymous Ref. 2
The reviewers' general point about relating these rates to biomass instead of volume was important and we changed the text in the abstract (L19) and the conclusions (L510) accordingly.
Specific Points
1. Added SD estimates for the per unit biomass estimates (L161)
2. We added text to explaining that we used log-transformations prior to statistical analyses when variance was heteroscedastic. L243
3. We now present the error measurement for the rates of nitrogen transformations per square meter, L366, based on our use of error estimates edited L161.
4. The values using mussel shells only are indeed presented separately from the whole mussel. That was intentional and in response to a previous review so that the reader can see the distinction.

Anonymous Ref 3
We thank the reviewer for the compliments.
The reviewer made an excellent point about better embedding our results in the context of the literature. We agree that it would provide a 'benchmark' and have enhanced section 4.1 (new paragraph starting L403), to show citations of nitrogen transformations in other systems, including the relatively low rates seen in open water systems and the higher rates in association with macrofauna.

We have addressed the minor technical problems with the manuscript that the reviewer mentioned and have made those edits to headings, parentheses.

[revised manuscript text omitted]

---

## Author Response (AR3)

*Dear Dr. Woulds:*
*I thank you for your careful edits. We have made the requested changes and our specific replies are below. We submit a 'track changes' version so you can see the edits that were made.*

**Associate Editor Decision: Publish subject to minor revisions (review by editor)** (28 Nov 2018) by Clare Woulds
Comments to the Author:
Dear DR Pfister

Thank you for revising your manuscript. I am happy with the changes to the discussion, but I am still not happy with the way that the data are presented. Please could you make the following changes before I make a final decision on your manuscript:

1) The 'rates' that you have added to the abstract and conclusions are in fact not rates, as you have not specified the timescale over which the stated amounts of N transformation occurred. Please add a temporal element to these values (i.e. report them in nmol per g wet biomass per h).

*This was an oversight when we were revising and we thank you for noticing. The units (either day or hour) have now been inserted in the abstract and conclusions.*

2) The rates that have been added to the abstract and conclusions should also be reported and discussed in the results and discussion sections (respectively). Consider using a tables(s), and/or additional figures. The abstract and conclusion sections should only contain a summary of findings presented in the main body of the paper.

*A table was a great solution to organizing these different rates. Table 1 now has these and we cite it when needed.*

3) The presentation of your figures does not yet meet publication standards. Firstly, the axis labels sometimes lack units (there are examples of this in all figures - the label 'ammonium oxidation' or 'nitrate reduction' is insufficient, it is necessary to provide units (nmol per L?). If you are just providing a change in concentration over the course of your incubation, then please make sure that the duration of the incubation is at least stated in the caption. Alternately, consider presenting all the data as rates (i.e. normalised to time as well).

*The figures have been revised with all of these comments in mind and all changes in nutrients are now listed as a rate per hour to be consistent with the ammonium oxidation and nitrate reduction rates.*

3) The visual presentation of figures 1 and 4 needs to be improved. In figure 1 I would discourage the use of colours, but consider making the data point markers larger. Please also tidy the legend so that it does not overlap the edge of the plot. In figure 4, the labels 'seawater', bioballs' and 'prionitis' could be given just once at the top of each stack of figures. This would make the figure less cluttered. If you had light greyscale shading in vertical zones with the 'seawater', bioballs' and 'prionitis' labels at the top, you could also get rid of colours in this figure. Please also enlarge this figure, so that the y axes are not so compressed.

*The figures were also revised making all the suggested changes, though the page size again added some constraints.*

*Again, many thanks and we hope you approve of the changes.*
*Catherine Pfister and Mark Altabet*

[revised manuscript text omitted]